# Effects of Maize–Crop Rotation on Soil Physicochemical Properties, Enzyme Activities, Microbial Biomass and Microbial Community Structure in Southwest China

**DOI:** 10.3390/microorganisms11112621

**Published:** 2023-10-24

**Authors:** Puchang Wang, Wenhui Xie, Leilei Ding, Yingping Zhuo, Yang Gao, Junqin Li, Lili Zhao

**Affiliations:** 1School of Life Sciences, Guizhou Normal University, Guiyang 550025, China; wangpuchang@163.com (P.W.); gyphoebe945@126.com (Y.G.); lijq489@nenu.edu.cn (J.L.); 2College of Animal Science, Guizhou University, Guiyang 550025, China; wenhui@nwafu.edu.cn (W.X.); gzzyzyp2023@163.com (Y.Z.); 3Guizhou Institute of Prataculture, Guiyang 550006, China; peterding2007gy@163.com

**Keywords:** rotation patterns, physicochemical properties, microbial biomass, enzyme activity, microbial communities

## Abstract

Introducing cover crops into maize rotation systems is widely practiced to increase crop productivity and achieve sustainable agricultural development, yet the potential for crop rotational diversity to contribute to environmental benefits in soils remains uncertain. Here, we investigated the effects of different crop rotation patterns on the physicochemical properties, enzyme activities, microbial biomass and microbial communities in soils from field experiments. Crop rotation patterns included (i) pure maize monoculture (CC), (ii) maize–garlic (CG), (iii) maize–rape (CR) and (iv) maize–annual ryegrass for one year (Cir1), two years (Cir2) and three years (Cir3). Our results showed that soil physicochemical properties varied in all rotation patterns, with higher total and available phosphorus concentrations in CG and CR and lower soil organic carbon and total nitrogen concentrations in the maize–ryegrass rotations compared to CC. Specifically, soil fertility was ranked as CG > Cir2 > CR > Cir3 > CC > Cir1. CG decreased enzyme activities but enhanced microbial biomass. Cir2 decreased carbon (C) and nitrogen (N) acquiring enzyme activities and soil microbial C and N concentrations, but increased phosphorus (P) acquiring enzyme activities and microbial biomass P concentrations compared to CC. Soil bacterial and fungal diversity (Shannon index) were lower in CG and Cir2 compared to CC, while the richness (Chao1 index) was lower in CG, CR, Cir1 and Cir2. Most maize rotations notably augmented the relative abundance of soil bacteria, including Chloroflexi, Gemmatimonadetes and Rokubacteria, while not necessarily decreasing the abundance of soil fungi like Basidiomycota, Mortierellomycota and Anthophyta. Redundancy analysis indicated that nitrate-N, ammonium-N and microbial biomass N concentrations had a large impact on soil bacterial communities, whereas nitrate-N and ammonium-N, available P, soil organic C and microbial biomass C concentrations had a greater effect on soil fungal communities. In conclusion, maize rotations with garlic, rape and ryegrass distinctly modify soil properties and microbial compositions. Thus, we advocate for garlic and annual ryegrass as maize cover crops and recommend a two-year rotation for perennial ryegrass in Southwest China.

## 1. Introduction

Maize (*Zea mays* L.) is the most significant cereal crop for food and industrial use worldwide, and it is mainly produced in the United States, China and Brazil [1,2]. In China, the North China Plain is a major maize-producing area, where not only did maize grain yields account for one-third of the national output from 2000 to 2019, but also where local farmers achieve higher returns from maize than from other crops [3,4]. Conversely, maize growers in the subtropical mountains of Southwest China face higher production costs and lower returns from maize cultivation due to the limitations of the dense population and the mountainous terrain, although maize is still the preferred and dominant crop for farmers in Southwest China. More diversified cropping systems can increase overall system productivity, which may compensate for cash loss from one of the crop species [5,6,7]. In the pursuit of higher economic benefits, smallholder farmers often choose different kinds of economic and forage crops as part of rotation with maize [8,9,10]. Therefore, adopting sustainable, effective rotation practices for increasing crop production and an improved economic perspective through the adoption of sustainable intensive farming systems in densely populated smallholder farming systems may be an important option.

Crop rotation has been recognized as an important on-farm management tool that can increase crop yields, reduce excessive use of chemical fertilizers and improve soil physicochemical properties [4,11,12,13,14]. For example, due to the well-developed rooting action of perennial forage grasses, the organic matter content of some farmland in maize–forage rotations was significantly higher than under monoculture maize cropping patterns [15,16]. Crop rotation can effectively improve soil nutrient content and enzyme activity [17,18,19] and change the diversity and structure of soil microbial communities [20,21] through the impacts of root exudates and crop residues, which are effective at reducing crop succession barriers [22]. One relevant study suggested that bacteria and fungi in soils generally account for more than 90% of all the soil microbial population [23], and some organic matter conversion processes in soils are controlled by soil microbial communities [24,25,26]. Soil microbial diversity can be used as an index for evaluating soil quality [26,27] and can be considered an important indicator for assessing soil stability under various environmental influences [28]. One reason is that diverse crop rotations affect the diversity and structure of the soil microbial community through root exudates, plant residues and symbiotic relationships [20,21]. Another reason is that multiple crop rotations can directly affect soil microbial community dynamics by changing soil carbon input, nutrient availability and soil structure [13,29]. Thus, exploring potential maize rotation patterns is extremely important for improving productivity and increasing the income of maize farmers in Southwest China.

Based on the above, local smallholder farmers often choose different economic crops or forage as part of a rotation with maize [7,9]. They believe that their crop rotation patterns are better for soil fertility, soil function and nutrient cycling [25]. However, it is not clear which of these maize-based cropping strategies is the most effective way of supporting sustainable intensification on smallholder farms. Meanwhile, there is conflicting evidence on the relative advantages of rotations over monocultures in improving soil fertility and supporting environmental sustainability [30,31]. Therefore, there is a need to understand the dynamics of soil physical, chemical and biological indicators under different maize–crop rotation patterns to evaluate and develop sustainable land-use options. The purposes of the current study were: (a) to determine the effects of different maize rotation patterns on soil physicochemical properties, enzyme activity, microbial biomass and microbial community structure and diversity and (b) to explore the mechanisms driving soil microbial community structure effects on soil nutrient concentrations under different rotation patterns and determine the major environmental factors that affect soil microbial community structure.

## 2. Materials and Methods

### 2.1. Study Site

The field experiment was situated in Fenggang County, Zunyi City, Guizhou Province, Southwest China (27°43′ N, 107°39′ E, elevation 826.3 m; Figure 1). This area has a subtropical monsoon climate. The average annual precipitation is 1559 mm, with more than 85% of the rainfall occurring between April and October. The average annual temperature is 14.8 °C, with the lowest temperature being −3 °C in January and the highest temperature being 35 °C in July. The soil is mainly yellow loam (Chinese soil taxonomy), which is weakly acidic [32]. In the field, the soil moisture content was 20–27%, and the concentrations of soil organic carbon (SOC), nitrate-nitrogen (NO_3_^−^-N), ammonium-nitrogen (NH_4_^+^-N) and available phosphorus (AP) were 19.5 g kg^−1^, 14.5 mg kg^−1^, 10.6 mg kg^−1^ and 17.2 mg kg^−1^, respectively.

### 2.2. Experimental Design and Soil Sampling

The experiment had a completely random block design with four replicate plots (150 m^2^ per replicate plot). The impact of crop rotation systems on soil properties was determined. Plots were separated into two regions where forage and economic crops were planted, including (i) pure maize (*Zea mays*) monoculture (CC), (ii) maize–garlic (*Allium sativum*) rotation (CG), (iii) maize–rape (*Brassica napus*) rotation (CR) and (iv) maize–forage (*Lolium multiflorum* L.) rotation consisting of one-year crop rotation (Cir1), two-year crop rotation (Cir2) and three-year crop rotation (Cir3). All crop rotation treatments were sown in plowed soil and formed 6 treatments in total. Each treatment had four replicates. The plot size was 10 × 10 m. To ensure that all treatments were sampled at the same time, sowing and harvesting times varied from October 2016 to September 2019, as shown in Table 1.

The plots of maize (cultivar Jinyu 838) had a spacing of 50 cm between rows and 25 cm among plants within a row [33]. Ryegrass (cultivar Bangde) was sown at a density of 75 kg ha^−1^. Garlic (local purple garlic) was planted at a spacing of 20 cm between rows and 12 cm among plants within a row. The oilseed rape (cultivar Youyan 57) was sown at a spacing of 60 cm between rows and 28 cm among plants within a row [34]. The N fertilizer was applied in the winter at 150 kg N ha^−1^ (forage/vegetable crops) and at 170 kg N ha^−1^ for maize. Each plot was fertilized with one-third of the nitrogen fertilizer being applied in the seedbed, with the remaining two-thirds being applied as a single top-dressing split at the jointing stage. P fertilizer was applied as a basic fertilizer at 60 kg P_2_O_5_ ha^−1^ yr^−1^ for both forage/economic crops and maize, whereas the application rates of K fertilizer were 40 and 50 kg K_2_O ha^−1^ yr^−1^ for forage/economic crops and maize, respectively.

In September 2019, five sampling points were set up in each replicate plot (using the diagonal five-point sampling method), and soil samples were collected during crop harvest in each sample plot [35]. Soil samples were collected from the rhizosphere zone of the crop at a depth of 0 to 20 cm and bulked into a single soil sample in each plot [19]. A total of 24 soil samples were obtained (six treatments × four replicates for each treatment). Each sample was divided into three subsamples, one of which was stored at −80 °C before being used for DNA extraction and analysis for microbial community analysis [36]. The second subsample was stored at 4 °C and used to measure microbial biomass and enzymatic activity [36]. The remaining subsample was air dried at 25–35 °C and used for the determination of soil physicochemical properties [36].

### 2.3. Soil Physicochemical Property Analysis

SOC was determined by the potassium dichromate volumetric method, NO_3_^−^-N by colorimetry and NH_4_^+^-N by ultraviolet spectrophotometry. The AP concentration was determined by the NaHCO_3_-ultraviolet spectrometry method [35], soil pH using a meter [19], total nitrogen (TN) by the Kjeldahl method [19] and total phosphorus (TP) by the alkali fusion molybdenum–antimony colorimetric method [36]. Soil water content (*SWC*) was determined by drying the soil sample; points were randomly selected in the quadrat, soil was taken using a ring knife, weighed (*M*_0_), sealed, brought back to the laboratory and dried to constant weight (*M*_1_) at 105 °C.
(1)$$SWC=\frac{M_{0}-M_{1}}{M_{1}}$$

Based on SOC, TN, TP, AP, NO_3_^−^-N and NH_4_^+^-N, a comprehensive evaluation measure of affiliated functions for soil fertility was obtained. The equations were as follows.
μ (*X_j_*) = (*X_j_* − *X_min_*)/(*X_max_* − *X_min_*)   *j* = 1, 2, 3, …, n(2)
where μ (*X_j_*) denotes the affiliation function value of the *j*th composite indicator, *X_j_* denotes the value of the *j*th composite indicator, *X_min_* denotes the minimum value of the *j*th composite indicator and *X_max_* denotes the maximum value of the *j*th composite indicator.

### 2.4. Soil Enzyme Activity and Microbial Biomass Analysis

We used an ELISA^®^ kit (Shanghai Enzyme-Linked Biotechnology Co., Ltd., Shanghai, China) [36] to detect the amount of soil β-1,4-glucosidase (BG), β-N-acetylglucosaminidase (NAG), leucine aminopeptidase (LAP), nitrogenase (Nit), nitric oxide synthase (NOS), glutamine synthetase (GS) and acid phosphatase (ACP), following the manufacturer’s instructions. Soil microbial biomass carbon (MBC) was measured by the CHCl_3_ fumigation–potassium dichromate volumetric method [19], whereas microbial biomass nitrogen (MBN) was determined by CHCl_3_ fumigation–distillation–HCl titration [19]. Microbial biomass phosphorus (MBP) was determined by the CHCl3 fumigation–total phosphorus method [19].

### 2.5. Soil Bacterial and Fungal DNA Extraction and Sequencing

Microbial genomic DNA was extracted from soil samples using the HiPure Soil DNA Kits (Magen, Guangzhou, China) in accordance with the manufacturer’s protocols. The V3 + V4 region of the 16S rDNA sequence was amplified from the genomic DNA using specific primer barcodes. The bacterial primer sequences were 341F (CCTACGGGNGGCWGCAG) and 806R (GGACTACHVGGGTATCTAAT) [19]. The ITS2 region of the ITS rDNA sequence was amplified from the genomic DNA. The fungal primer sequences were ITS3_KYO2 (GATGAAGAACGYAGYRAA) and ITS4 (TCCTCCGCTTATTGATATGC). The detailed conditions for PCR and amplicon sequencing were described in our recent study [19]. PCR reactions were performed in triplicate using an Applied Biosystems ProFlex 2 × 96-well PCR instrument (9902, ABI, New York, NY, USA). The sequencing was performed in Hiseq2500 Illumina system with PE250 mode by Guangzhou Genedenovo Co., Ltd., Guangzhou, China. 

The UPARSE pipeline was used to cluster valid tags into operational taxonomic units (OTUs) with ≥97% similarity [19,37]. The Chao1 and Shannon indices were calculated in QIIME. For the sake of convenience, here, we present only the top ten species with expression abundance of 2% in at least one sample, whereas the remainder of the species are categorized as “other”, and tags that cannot be annotated to that level are categorized as “not classified”. Sequence alignment was carried out by using Muscle [38] (version 3.8.31) (http://www.drive5.com/muscle/, accessed on 1 January 2004). Redundancy analysis (RDA) was performed in R (R Development Core Team, 2018) [35].

### 2.6. Statistical Analysis

Soil pH, WC, SOC, TN, TP, NO_3_^−^-N, NH_4_^+^-N, AP, BG, NAG, LAP, Nit, NOS, GS, ACP, MBC, MBN and MBP, as well the Shannon index and Chao1 index, were analyzed by SPSS (version 22.0) (IBM, Armonk, NY, USA). When the F-test from analysis of variance was significant (*p* < 0.05), Tukey’s test was used for multiple pairwise comparison of samples. Pearson’s linear correlation coefficients were calculated for all the physicochemical and microbiological attributes of the soil [36]. 

## 3. Results

### 3.1. Soil Physical and Chemical Properties

Compared with the maize monoculture pattern, the maize–garlic, maize–rape and forage crop rotations significantly altered the physicochemical properties of the rhizosphere soil (Table 2). Soil WC increased markedly under the maize–garlic, maize–rape and forage rotations relative to the monoculture, CC, whereas there was no consistent effects on soil pH. In addition, soil nutrient concentration showed that there was higher NH_4_^+^-N, TP and AP in maize–garlic and maize–rape rotations but lower SOC and T concentrations in the maize–forage crop rotations compared to that in the maize monoculture pattern. Specifically, CG and Cir2 had higher AP, NO_3_^−^-N and NH_4_^+^-N compared to the CC pattern. Moreover, our study carried out a comprehensive evaluation of soil fertility values based on the affiliation function method (Table 3), and the results showed that the soil fertility of the six cropping patterns was in the order CG > Cir2 > CR > Cir3 > CC > Cir1.

### 3.2. Soil Enzyme Activity

In terms of soil enzyme activities (Figure 2), different maize–garlic, maize–rape and forage rotations led to significant changes in BG, NOS, Nit, LAP and ACP activities (*p* < 0.05), but no significant change was observed for NAG or GS activities (*p* > 0.05). Specifically, BG activity was lower in the CG, Cir2 and Cir3 patterns than in the CC pattern, although the opposite was observed in the Cir1 and CR patterns. NOS and ACP activities were lower in the CG, CR, Cir1 and Cir3 patterns than in the CC pattern, whereas the levels of those enzymes were higher in the Cir2 pattern than in the CC pattern. Nit was lower in the CG and CR patterns than in the CC pattern, but the opposite was observed in Cir1, Cir2 and Cir3.

### 3.3. Soil Microbial Biomass

For soil microbial biomass (Figure 3), different maize–garlic, maize–rape and forage rotations led to significant variations in MBC, MBN and MBP (*p* < 0.05). MBC and MBN concentrations were significantly higher in CG and Cir3 than in the CC pattern, while they were lower in CR, Cir1 and Cir2 patterns than in the CC pattern. MBP was significantly higher in CG, CR and Cir2 patterns than in the CC pattern, although the opposite trend was observed in the Cir1 and Cir3 patterns.

### 3.4. Microbial Community Diversity and Structure

A total of 2,234,017 bacterial sequences and 2,366,741 fungal sequences were obtained. High-quality sequences constituted 91.26% of the original bacterial sequences and 91.09% of the original fungal sequences (Appendix A). These proportions suggest that the sampling and sequencing depths were satisfactory. Additionally, for bacterial OTUs, the ranges obtained were 3978–4947 in CC, 3909–4600 in CG, 4624–5560 in CR, 3569–4775 in Cir1, 2388–4071 in Cir2 and 4566–5870 in Cir3 (Appendix A). For fungal OTUs, the ranges were 766–895 in CC, 707–826 in CG, 746–817 in CR, 725–766 in Cir1, 644–804 in Cir2 and 695–989 in Cir3 (Appendix A). There were marked differences in soil microbial community diversity (Shannon index) and richness (Chao1 index) under different maize–garlic, maize–rape and forage crop rotations (Table 4). Soil bacteria Shannon index was lower in the CG and Cir2 patterns than in the CC pattern, whereas Chao1 index was lower in the CG, CR, Cir1 and Cir2 patterns than in the CC pattern. The variations in fungal community diversity and richness were similar to those in soil bacteria under the different crop rotations.

In terms of phylum level, the community structures of soil bacterial and fungal communities differed sharply among the six cropping patterns. The results indicated that Proteobacteria was the most abundant bacterial phylum and accounted for 26.35% of the total sequences, with the remaining phyla Chloroflexi, Planctomycetes, Acidobacteria, Actinobacteria and Gemmatimonadetes having relative abundances of 12.98%, 10.52%, 15.95%, 10.69% and 10.24% of the total sequences, respectively (Figure 4). Compared with the CC pattern, the relative abundances of Proteobacteria, Actinobacteria and Bacteroidetes were higher in the Cir2 pattern, whereas those of Chloroflexi, Gemmatimonadetes and Rokubacteria were higher in the CG, Cir2 and CR patterns. With respect to soil fungal communities, the relative abundances of Basidiomycota, Mortierellomycota and Anthophyta were lower in the Cir2 and CG patterns than in the CC pattern (Figure 5). Moreover, this study found that the average abundance of unclassified fungal species was higher than those of bacteria among the six cropping patterns. At the genus level, the top 10 bacterial genera in terms of relative abundance are Gemmatimonas (1.87–8.31%), Sphingomonas (0.82–2.91%), Streptomyces (0.12–6.69%), Rhodanobacter (0.30–5.45%), Chujaibacter (0.31–3.12%), Acidothermus (0.10–2.22%), Haliangium (0.25–2.26%), HSB_OF53-F07 (0.02–2.65%), Candidatus Ssolibacter (0.65–1.74%) and Bryobacter (0.75–1.49%) (Appendix A). Similarly, the top 10 fungal genera, which coexist in all treatments but with varying relative abundances, include Penicillium (1.37–24.54%), Mortierella (1.25–7.83%), Fusarium (0.33–5.76%), Setophoma (0.15–2.59%), Typhula (0.00–7.31%), Auricularia (0.01–4.05%), Plectosphaerella (0.13–4.27%), Codinaea (0.01–3.58%), Aspergillus (0.08–2.34%) and Arachis (0.06–3.51%) (Appendix A). 

### 3.5. Correlation of Soil Physical and Chemical Properties with Microbial Taxa Parameters

Pearson’s correlation analysis indicated that bacterial community diversity was significantly positively correlated with soil WC, pH and SOC and significantly negatively correlated with NO_3_^−^-N and NH_4_^+^-N (Table 5). Bacterial RDA indicated that the first axis (RDA1) accounted for 35.24% of the total bacterial phylum variation and was significantly correlated with WC, MBN, NH_4_^+^-N and NO_3_^−^-N, whereas the second axis (RDA2) accounted for 27.02% of the whole bacterial phylum variation, and pH and SOC were closely correlated with RDA2. The two axes together explained 62.26% of the total variation in the bacterial community composition (Figure 6a). Proteobacteria abundance was positively correlated with NO_3_^−^-N, NH_4_^+^-N and pH, but negatively correlated with WC, MBN, MBC and SOC. Acidobacteria abundance was positively correlated with NO_3_^−^-N and NH_4_^+^-N but negatively correlated with pH, WC, MBN, MBC and SOC. Planctomycetes, Actinobacteria and Gemmatimonadetes abundances were positively correlated with pH, WC, MBN, MBC and SOC.

Pearson’s correlation analysis indicated that the fungal community diversity index was significantly and negatively correlated with NH_4_^+^-N content (Table 4). RDA indicated that the first axis (RDA1) accounted for 31.34% of the fungal phylum variation and was closely related to soil pH, AP and TP concentrations, and the second axis (RDA2) explained 13.27% of the fungal phylum variation and was closely related to NO_3_^−^-N, NH_4_^+^-N, MBC and SOC (Figure 6b). Ascomycota abundance was positively correlated with pH, NO_3_^−^-N and NH_4_^+^-N and negatively correlated with AP, TP, SOC, MBC and MBN. Basidiomycota abundance was positively correlated with AP, TP, SOC and MBC and negatively correlated with NO_3_^−^-N, NH_4_^+^-N and pH. Chlorophyta and Anthophyta abundances were positively correlated with pH, MBN and MBC and negatively correlated with NO_3_^−^-N and NH_4_^+^-N. 

Therefore, RDA and Pearson’s correlation analysis indicated that NO_3_^−^-N, NH_4_^+^-N, MBN, AP, SOC and MBC were the main environmental factors which affected bacterial and fungal community structures. Moreover, soil WC, pH, SOC, NO_3_^−^-N and NH_4_^+^-N concentrations sharply influenced the microbial community diversity (Table 5).

## 4. Discussion

In this study, the maize–economic crop and maize–forage crop rotation patterns had higher soil WC than the pure maize monoculture pattern (Table 2). One possible explanation is that the economic crop or forage grow rapidly and luxuriantly with a large canopy cover (behaving as a cover crop) in the winter season, which inhibits soil water evaporation and hence increases soil WC in the plough layer [39]. Another reason is that the well-developed fibrous roots of an economic crop or forage will form a stable agglomerate structure in the soil [40], which improves soil erosion resistance and slows and reduces surface runoff [41]. SOC and TN concentrations were higher in pure maize monoculture pattern than in maize–economic crop and maize–forage rotation patterns due to the previous long-term maize monoculture, and almost all of the post-harvest maize stover was returned to the field. Maize stover is rich in organic carbon and returning it to the field increased the SOC content [42]. Many studies have indicated that straw return practices may maintain or even improve soil TN content in semi-arid cropping systems compared to conventional tillage [43,44,45]. Our results suggest that this conclusion also holds true in Southwest China. Compared to monoculture, rotation has been shown to be an effective cultivation strategy for preventing soil N loss [46,47,48]. Since NH_4_^+^-N is positively charged, it is easily adsorbed onto the surface soil and enrichment occurs at the soil surface, making the NH_4_^+^-N less capable of migrating to the lower soil layers [49]. Therefore, maize–economic crop and maize–forage rotation patterns can significantly slow down the runoff rate to reduce surface runoff, which may be an important reason for the significantly higher soil NH_4_^+^-N concentration in this study. 

Soil P deficiency in agricultural soils is a common problem in Southwest China [6]. Although P is one of the major nutrients required for plant cultivation, not all P exists in a soluble form and only soil AP is directly absorbed and used by plants [50]. Our study showed that the maize–garlic, maize–rape and the maize–forage crop rotation patterns markedly increased the soil TP and AP concentrations (Table 2). In general, the plant root system can improve soil structure, activate the use of insoluble P in the soil, transfer nutrients from the deeper soil layers to the surface zone and release P during the decomposition of organic matter [51]. Therefore, the maize–garlic and maize–rape rotation patterns were effective in increasing soil P concentration and improving soil fertility compared to maize monoculture pattern. These results suggest that maize–garlic, maize–rape and maize–forage crop rotations can effectively improve certain physicochemical properties of agricultural soils [52]. Moreover, our study determined a comprehensive evaluation value for soil fertility with ranking in the order: CG > Cir2 > CR > Cir3 > CC > Cir1.

Rotation practices affect soil properties by increasing plant diversity. In this way, it alters enzyme activity and microbial biomass. Maize–garlic, maize–rape and maize–forage crop rotations resulted in marked differences in soil enzyme activities and SMB, and these variation trends were in accordance with variation in the concentrations of most soil nutrients. BG itself, as an important class of hydrolytic enzyme in soil, can reflect the intensity of heterotrophic respiration using organic carbon as a substrate in the soil [53]. Our study found there was lower BG activity in the CG, Cir2 and Cir3 patterns, compared to the CC pattern, a finding which is consistent with the SOC change trends. Nitric oxide synthase (NOS) is a common enzyme which is responsible for NO synthesis in bacterial cells [54], which is essential for plants because NO produced by soil bacteria can serve as an additional source of nitrogen for plants [55]. Studies have indicated that NOS is higher in bacteria of the Actinobacteria Gram-positive phylum [56], which could be the reason for the higher NOS activity in Cir2. Nitrogenase is an enzyme that can reduce gaseous nitrogen molecules to ammonia, which is available to plants [57]. In our study, the concentration of soil NH_4_^+^-N had a cumulative effect with the prolongation of maize–ryegrass rotation, resulting in significant increases in Nit activity in the Cir1, Cir2 and Cir3 patterns. The reason may be that sowing ryegrass caused the soil to form a stable agglomerate structure, resulting in greater soil porosity and better aeration, which enhanced the activity of the soil nitrogen fixation enzyme. ACP plays a significant role in hydrolyzing soil organic phosphorus and increasing soil phosphorus utilization by plants [58]. The gene expression of root ACP and its enzyme activity increased under phosphorus deficiency conditions in plants [59]. In the present study, there was lower TP in the Cir2, Cir3 and CC patterns and higher ACP in the Cir2, Cir3 and CC patterns. LAP is a hydrolase involved in microbial acquisition of N [60], whereas other studies found that soil LAP activity was positively linked with SOC [61]. The higher SOC concentration in the CC pattern may be an important factor for the higher LAP activity in our study.

The total amount of biomass in the soil, excluding living plants with a volume of less than 5 × 10^3^ μm, is called the SMB and is considered to be one of the important indicators of variation in soil fertility [62,63]. Our study supported this view that the comprehensive evaluation of the soil nutrient content in CG was highest, and its MBC, MBN and MBP concentrations were also significantly higher than in the maize monoculture. The well-developed taproot system of rape improves soil physicochemical properties, resulting in greater soil biochemical activity and soil microbial activity, which significantly increases MBC, MBN and MBP concentrations [64]. SMB is significantly affected by rotation [65], but further research is needed on the mechanism by which soil rotation patterns affect soil MBC, MBN and MBP concentration changes. The results indicate that the maize–garlic, maize–rape and maize–forage rotation patterns have a positive effect on soil enzyme activity and microbial biomass, which reflects the trend and intensity of various biochemical processes, but also reflects increases in soil fertility and health status to some extent.

Our study indicated that bacterial community diversity was positively correlated with soil WC, pH and SOC concentration and negatively correlated with soil NH_4_^+^-N and NO_3_^−^-N concentrations. Soil bacterial community diversity was lower in the CG and Cir2 patterns than in the CC patterns, whereas the opposite association was observed in the CR, Cir1and Cir3 patterns. Unlike bacterial community diversity, fungal community diversity was negatively correlated with soil NH_4_^+^-N concentration. Meanwhile, the changing trend in fungal diversity was correlated with variation in the bacterial community diversity. This result indicates that the soil physicochemical properties and rotation practices play a vital role in regulating the characteristics of the microbial community structure. Previous studies have indicated that soil pH has a significant impact on soil bacterial diversity [66,67], which is consistent with the finding that high bacterial community diversity was positively correlated with soil pH in our study. Higher SWC in the maize–garlic, maize–rape rotation and maize–forage rotation patterns also had a significant effect on bacterial community diversity, a possible reason for this is that soil water concentration can affect soil nutrient transport, substrate availability and other soil properties, leading to changes in the community composition and activity of soil microorganisms [68]. SOC content has been reported to have an important impact on soil bacterial diversity [69], and, in our study, bacterial community composition was positively correlated with SOC content because it provides a carbon source for soil microorganisms [70]. Fungi have a greater requirement for nutrients in the environment. The increased crop growth in the maize–garlic, maize–rape rotation and maize–forage rotation patterns limits the growth of fungi because of nutrient shortages or secretion of allelochemicals, with similar values of the comprehensive evaluation measure of soil quality in our study. For instance, as garlic root systems inhibit certain extra-root microorganisms by releasing chemicals into the environment [71], garlic chemosensory effects can largely affect soil fungal community diversity by reducing the diversity and abundance of harmful soil fungi [72].

The composition of soil microbial communities in agricultural systems is complex, and rotation practices have a significant effect on the soil microorganisms [33,73,74]. Soil microbial community structure is also a significant indicator to test soil health and vitality [73,75]. Our study found that the relative abundance of dominant bacterial and fungal phyla in communities in soil is closely correlated to soil nutrient concentrations. The phylum with the highest relative abundance in bacterial communities is Proteobacteria, members of which are the main dominant taxa in terrestrial ecosystems with a wide ecological range and high environmental adaptability [76,77]. The relative abundance of Proteobacteria was positively correlated with soil TN and AP and closely related to soil fertility [78,79], all of which are consistent with the results in the current study. Meanwhile, our results found that the relative abundance of Acidobacteria in soil is second only to that of Proteobacteria, and its members are mainly acidophilic bacteria. Acidobacteria abundance is also an important microbial indicator of soil nutrient status [80]. Previous studies [81,82] showed that bacteria of the Acidobacteria phylum are oligotrophic, whereas the current study showed that the relative abundance of Acidobacteria in this study was positively correlated with NH_4_^+^-N, NO_3_^−^-N, AP, TP and TN concentrations, indicating that not all Acidobacteria members are oligotrophic [33]. CG and Cir2 patterns improved soil nitrogen supply and nutrient status, promoting the reproduction of Proteobacteria and Actinobacteria associated with nitrogen cycling and inhibiting the growth of Acidobacteria, results which are generally in agreement with the findings of Ligi et al. [83]. Chloroflexi is a group of parthenogenic anaerobic bacteria that mainly perform carbon and nitrogen fixation [84]. The results of the present study indicated that the relative abundance of Chloroflexi was negatively correlated with soil TN and pH, with the negative correlation possibly being related to soil N competition [79]. The relative abundance of Actinobacteria in CG and Cir2 soils was higher than in other treatments, and the soil pH of these treatments were only 5.22 and 4.76, respectively, indicating that Actinobacteria bacteria prefer to live in acidic soils [85]. Previous studies had shown that the relative abundance of Gemmatimonadetes was positively linked with soil MBN concentration and its members may actively participate in soil N cycling and sulfate reduction [79,86], and the relative abundance of Gemmatimonadetes is lower in acidic soils [87,88]. This finding is different from the positive correlation between the relative abundance of Gemmatimonadetes and both soil MBN concentration and pH in the current study. Differences in bacterial genera between treatments were observed (Appendix A), with potential implications for soil health. Genera such as Gemmatimonas and Sphingomonas are often associated with promoting plant growth [89], while Streptomyces enhances soil fertility [90]. Rhodanobacter has the capacity to tolerate low pH and offers protection against the root rot fungal pathogen Fusarium [91,92]. Chujaibacter is commonly found in healthy soils [93]. The genus Acidothermus thrives in acidic conditions and can degrade plant tissues [94], while Haliangium aids in phosphate solubilization and promotes plant growth [95]. Both HSB_OF53-F07 and Acidothermus genera play roles in enhancing soil carbon metabolism [96]. Candidatus Solibacter has capabilities for heavy metal soil remediation [97], and Bryobacter exhibits chemorganotrophic activity, utilizing sugars, polysaccharides and organic acids as energy sources [98]. Notably, the Cir2 pattern demonstrated the highest relative abundance of the Streptomyces, Rhodanobacter, Chujaibacter and Acidothermus genera (Appendix A), suggesting its potential benefits for plant health. Based on these findings, the Cir2 pattern is recommended. The dominant clades of soil fungi in all rotation patterns were Ascomycota and Basidiomycota, both of which had an average relative abundance of more than 50%; Ascomycota and Basidiomycota are generally identified as the two major fungal clades in soil [70,99]. In general, members of Ascomycota are soil saprophytic fungi that degrade soil organic matter, although some members of Ascomycota cause plant diseases, such as root rot, stem rot, fruit (cob) rot and branch blight. Additionally, the genera Fusarium, Setophoma and Plectosphaerella have been associated with plant rot [100,101], while Typhula is known to cause snow blight [102]. Conversely, genera such as Penicillium, Mortierella and Aspergillus function as plant growth-promoting fungi [102,103,104,105]. In our study, the Cir2 pattern displayed the highest relative abundance of the Penicillium genus and the lowest of Fusarium and Setophoma (Appendix A). This suggests that the Cir2 pattern could be advantageous for plant health. 

As mentioned above, garlic root secretions affect the soil fungal community and inhibit the growth of some phytopathogenic fungi. Basidiomycota had its highest relative abundance in the Cir2 and CG patterns. Soil organic matter is a source of carbon and nitrogen for microorganisms, and straw return and green manure return affect the decomposition process of crop residues, which promotes the accumulation of SOC. The abundance of Ascomycota was positively correlated with soil pH and concentrations of NH_4_^+^-N and NO_3_^−^-N, whereas the abundance of Basidiomycota was negatively correlated, showing that different types of fungi responded differently to soil pH and effective N concentration [106].

## 5. Conclusions

This study examined soil quality and microbial community structures in southwestern China under various rotation patterns: pure maize monoculture, maize–garlic, maize–rape and maize–forage of differing durations. Overall, the maize–garlic, maize–rape and maize–forage rotations significantly influenced soil physicochemical properties, enzyme activities, microbial biomass and the diversity and structure of the microbial community. Soil nutrient quality was ranked as CG > Cir2 > CR > Cir3 > CC > Cir1, exhibiting changes in microbial composition and enzyme activities. Simultaneously, the diversity of both bacterial and fungal communities displayed pronounced patterns, with notably reduced bacterial and fungal community diversity in the CG and Cir2 patterns compared to the CC pattern. Our findings empirically advocate for the incorporation of garlic and ryegrass into local maize rotation practices and recommend extending the ryegrass rotation period to two years in the subtropical mountainous regions of Southwest China.

## Figures and Tables

**Figure 1 microorganisms-11-02621-f001:**
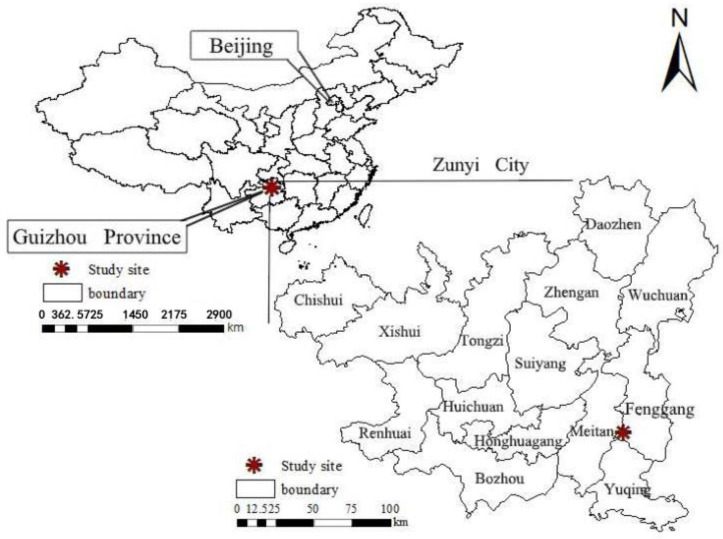
Location and basic information on experiment sites in this study.

**Figure 2 microorganisms-11-02621-f002:**
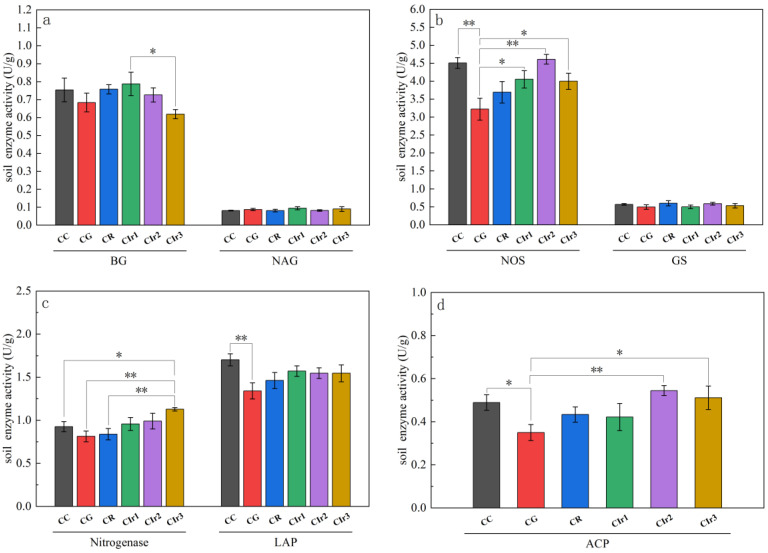
Soil enzyme activity of BG and NAG (**a**), NOS and GS (**b**), nitrogenase and LAP (**c**) and ACP (**d**) for the six cropping patterns. Note: pure maize monoculture (CC), maize–garlic rotation (CG), maize–rape rotation (CR) and maize–annual ryegrass of one-year crop rotation (Cir1), two-year crop rotation (Cir2) and three-year crop rotation (Cir3). BG: β-1,4-glucosidase; NAG: β-*N*-acetylglucosaminidase; NOS: nitric oxide synthase; GS: glutamine synthetase; LAP: leucine aminopeptidase; ACP: acid phosphatase. Values are the means ± standard errors. * Indicates a significant difference at 0.05 level. ** Indicates a significant difference at 0.01 level.

**Figure 3 microorganisms-11-02621-f003:**
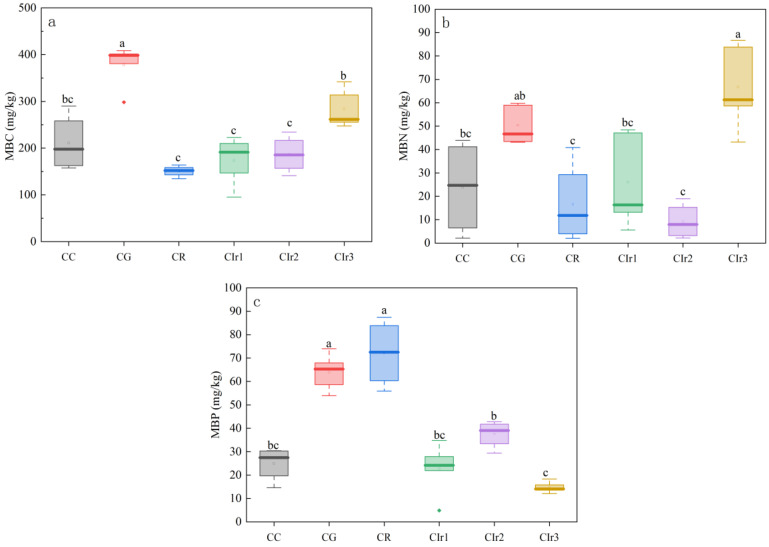
Soil MBC (**a**), MBN (**b**) and MBP (**c**) for the six cropping patterns. Note: pure maize monoculture (CC), maize–garlic rotation (CG), maize–rape rotation (CR) and maize–annual ryegrass of one-year crop rotation (Cir1), two-year crop rotation (Cir2) and three-year crop rotation (Cir3). MBC: microbial biomass carbon. MBN: microbial biomass nitrogen. MBP: microbial biomass phosphorus. Values are the means ± standard errors. Different letters within a figure show significant differences (*p* < 0.05) based on analysis of variance and Tukey’s test for pairwise comparisons. The red rhombus is discrete point.

**Figure 4 microorganisms-11-02621-f004:**
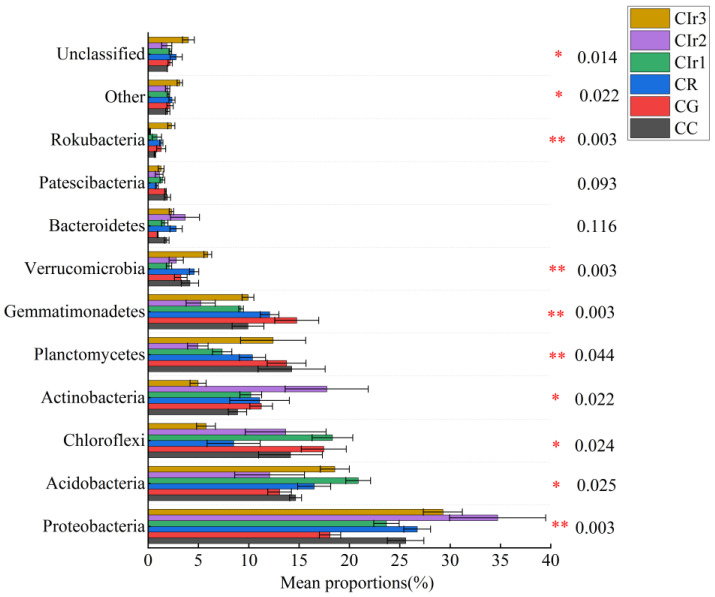
Relative abundance (%) of major soil bacterial phyla under six rotation cropping patterns (relative abundance ≥ 1% in at least one sample). Note: pure maize monoculture (CC), maize–garlic rotation (CG), maize–rape rotation (CR) and maize–annual ryegrass of one-year crop rotation (Cir1), two-year crop rotation (Cir2) and three-year crop rotation (Cir3). The vertical axis represents the respective phyla, the column length represents the average relative abundance of the corresponding phylum in each sample group and the different colors represent different cropping patterns. The numbers on the right-hand side are the *p* values. * Indicates a significant difference between the cropping patterns at 0.05 level. ** Indicates a significant difference at 0.01 level.

**Figure 5 microorganisms-11-02621-f005:**
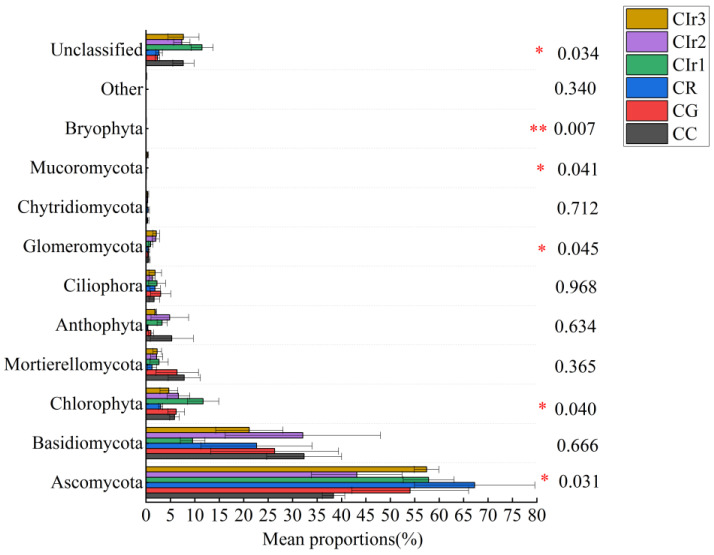
Relative abundance (%) of major soil fungal phyla under six rotation cropping patterns (relative abundance ≥ 1% in at least one sample). Note: pure maize monoculture (CC), maize–garlic rotation (CG), maize–rape rotation (CR) and maize–annual ryegrass of one-year crop rotation (Cir1), two-year crop rotation (Cir2) and three-year crop rotation (Cir3). The vertical axis represents the phylum, the column length represents the average relative abundance of the corresponding phylum in each sample group and the different colors represent different cropping patterns. The numbers at the right-hand side are the *p* values. * Indicates a significant difference between cropping patterns at 0.05 level. ** Indicates a significant difference at 0.01 level.

**Figure 6 microorganisms-11-02621-f006:**
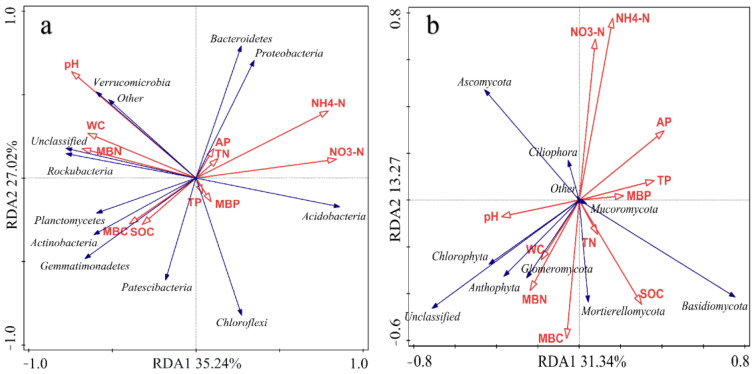
Redundancy of bacterial (**a**) and fungal (**b**) communities at phylum level and soil properties for six cropping patterns. The red arrows represent the different soil physical and chemical properties and the blue arrows represent the top ten bacterial (**a**) or fungal (**b**) phyla in terms of abundance.

**Table 1 microorganisms-11-02621-t001:** The sowing and harvesting times of the crop rotations.

Treatments	Plant	One Rotation		Two Rotations		Three Rotations	
		Sowing Date	Harvest Date	Sowing Date	Harvest Date	Sowing Date	Harvest Date
CC	*Zea mays*	May 2017	September 2017	May 2018	September 2018	May 2019	September 2019
CG	*Allium sativum*	---	---	---	---	October 2018	April 2019
	*Zea mays*	---	---	---	---	May 2019	September 2019
CR	*Brassica napus*	---	---	---	---	October 2018	April 2019
	*Zea mays*	---	---	---	---	May 2019	September 2019
Cir1	*Lolium multiflorum*	---	---	---	---	October 2018	April 2019
	*Zea mays*	---	---	---	---	May 2019	September 2019
Cir2	*Lolium multiflorum*	---	---	October 2017	April 2018	October 2018	April 2019
	*Zea mays*	---	---	May 2018	September 2018	May 2019	September 2019
Cir3	*Lolium multiflorum*	October 2016	April 2017	October 2017	April 2018	October 2018	April 2019
	*Zea mays*	May 2017	September 2017	May 2018	September 2018	May 2019	September 2019

Note: ‘---‘ represents the crop rotation test that has not started in this plot, which belongs to maize monoculture. Pure maize monoculture (CC), maize–garlic rotation (CG), maize–rape rotation (CR) and maize–annual ryegrass of one-year crop rotation (Cir1), two-year crop rotation (Cir2) and three-year crop rotation (Cir3).

**Table 2 microorganisms-11-02621-t002:** Soil physical and chemical properties for six cropping patterns.

Indicator	Treatments						*p*-Value
	CC	CG	Cr	CIr1	CIr2	CIr3	
WC (%)	21.38 ± 1.22 b	22.82 ± 1.00 ab	23.22 ± 1.00 ab	20.05 ± 1.00 b	24.01 ± 2.08 ab	26.23 ± 1.00 a	<0.05
pH	5.32 ± 0.05 bc	5.22 ± 0.10 c	6.05 ± 0.31 a	5.22 ± 0.10 c	4.76 ± 0.11 c	6.04 ± 0.31 a	<0.05
SOC (g kg^−1^)	23.13 ± 0.66 a	19.74 ± 0.72 bc	21.91 ± 0.53 ab	18.93 ± 0.55 c	16.27 ± 0.96 d	17.59 ± 0.95 cd	<0.05
TN (g kg^−1^)	2.18 ± 0.04 a	1.96 ± 0.07 bc	2.00 ± 0.06 ab	1.74 ± 0.02 d	1.85 ± 0.09 bcd	1.78 ± 0.05 cd	<0.05
TP (g kg^−1^)	0.54 ± 0.02 d	0.77 ± 0.02 b	1.03 ± 0.04 a	0.79 ± 0.02 b	0.68 ± 0.01 c	0.56 ± 0.02 d	<0.05
AP (mg kg^−1^)	7.60 ± 1.48 c	10.83 ± 0.95 bc	49.73 ± 7.12 a	11.30 ± 2.49 bc	19.49 ± 0.98 b	4.17 ± 0.88 c	<0.05
NO_3_^−^-N (mg kg^−1^)	12.26 ± 2.42 b	15.35 ± 3.22 b	5.57 ± 4.78 c	15.93 ± 3.05 b	33.56 ± 4.42 a	4.59 ± 1.27 c	<0.05
NH_4_ ^+^-N (mg kg^−1^)	2.57 ± 0.27 c	4.29 ± 2.12 c	6.02 ± 1.92 c	11.32 ± 5.84 b	19.87 ± 4.93 a	19.37 ± 0.85 a	<0.05

Note: pure maize monoculture (CC), maize–garlic rotation (CG), maize–rape rotation (CR) and maize–annual ryegrass of one-year crop rotation (Cir1), two-year crop rotation (Cir2) and three-year crop rotation (Cir3). Values are the means ± standard errors of the mean of three plots. Within a row, samples with different letters denote significant differences (*p* < 0.05) based on analysis of variance and Tukey test for pairwise comparisons. SWC—soil water content; TN—total nitrogen; TP—total phosphorus; AP—available phosphorus; SOC—soil organic carbon; NO_3_^−^-N—nitrate nitrogen; NH_4_^+^-N—ammonium nitrogen.

**Table 3 microorganisms-11-02621-t003:** Combined scores of soil fertility indicators and their rankings for six cropping patterns.

Indicator	Treatments					
	CC	CG	Cr	CIr1	CIr2	CIr3
SOC	0.49	0.60	0.46	0.54	0.50	0.48
TN	0.50	0.56	0.53	0.49	0.43	0.56
TP	0.55	0.53	0.56	0.28	0.44	0.32
AP	0.44	0.46	0.41	0.34	0.56	0.49
NO_3_^−^-N	0.38	0.30	0.61	0.55	0.57	0.49
NH_4_^+^-N	0.34	0.58	0.3	0.43	0.39	0.44
Degree of membership	2.71	3.03	2.87	2.63	2.89	2.78
Ranking	5	1	3	6	2	4

Note: pure maize monoculture (CC), maize–garlic rotation (CG), maize–rape rotation (CR) and maize–annual ryegrass of one-year crop rotation (Cir1), two-year crop rotation (Cir2) and three-year crop rotation (Cir3).

**Table 4 microorganisms-11-02621-t004:** Soil bacterial and fungal richness (Chao1 index) and diversity (Shannon index) for the six cropping patterns.

Indicator		Treatments						*p*-Value
		CC	CG	CR	CIr1	CIr2	CIr3	
Bacterial	Shannon	10.16 ± 0.15 ab	9.57 ± 0.10 bc	10.38 ± 0.16 ab	9.81 ± 0.15 abc	9.16 ± 0.53 c	10.52 ± 0.19 a	0.013
	Chao1	6080 ± 224 a	5660 ± 153 a	4852 ± 1624 a	5056 ± 198 a	4354 ± 447 a	6401 ± 425 a	0.363
Fungi	Shannon	5.63 ± 1.01 a	5.20 ± 0.65 a	6.15 ± 0.42 a	6.56 ± 1.30 a	4.52 ± 0.51 a	6.25 ± 0.65 a	>0.05
	Chao1	1101 ± 33 a	1068 ± 72 ab	1044 ± 62 ab	970 ± 19 ab	954 ± 38 c	1069 ± 46 ab	0.046

Note: pure maize monoculture (CC), maize–garlic rotation (CG), maize–rape rotation (CR) and maize–annual ryegrass of one-year crop rotation (Cir1), two-year crop rotation (Cir2) and three-year crop rotation (Cir3). Different letters on samples within a row show significant differences (*p* < 0.05) based on analysis of variance and Tukey test pairwise comparisons.

**Table 5 microorganisms-11-02621-t005:** Correlations (*r*) between microbial community diversity (Shannon index), richness (Chao1 index) and soil physicochemical properties.

Indicator	Bacterial				Fungal			
	Chao 1		Shannon		Chao 1		Shannon	
	*r*	*p*	*r*	*p*	*r*	*p*	*r*	*p*
WC	0.365	*p* > 0.05	0.431 *	0.036	0.168	*p* > 0.05	0.236	*p* > 0.05
pH	0.508 *	0.011	0.526 **	0.008	0.102	*p* > 0.05	0.189	*p* > 0.05
SOC	0.498 *	0.013	0.446 *	0.029	−0.002	*p* > 0.05	0.143	*p* > 0.05
TN	0.124	*p* > 0.05	−0.089	*p* > 0.05	−0.053	*p* > 0.05	−0.103	*p* > 0.05
TP	0.151	*p* > 0.05	0.082	*p* > 0.05	−0.34	*p* > 0.05	−0.027	*p* > 0.05
AP	0.168	*p* > 0.05	0.095	*p* > 0.05	−0.397	*p* > 0.05	−0.105	*p* > 0.05
NO_3_^−^-N	−0.469 *	0.021	−0.581 **	0.003	−0.518 **	0.01	−0.283	*p* > 0.05
NH_4_^+^-N	−0.624 **	0.001	−0.591 **	0.002	−0.366	*p* > 0.05	−0.418 *	0.042
MBC	0.129	*p* > 0.05	−0.025	*p* > 0.05	0.214	*p* > 0.05	−0.075	*p* > 0.05
MBN	0.311	*p* > 0.05	0.256	*p* > 0.05	0.271	*p* > 0.05	0.018	*p* > 0.05
MBP	0.076	*p* > 0.05	−0.137	*p* > 0.05	−0.252	*p* > 0.05	−0.126	*p* > 0.05

Values are the means ± standard deviation SD of three biological replicates. * Indicates a significant correlation at 0.05 level, ** Indicates a significant correlation at 0.01 level.

## Data Availability

The original contributions presented in the study are included in the article. Further inquiries can be directed to the corresponding authors.

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
