# Peer review of "Effects of Maize–Crop Rotation on Soil Physicochemical Properties, Enzyme Activities, Microbial Biomass and Microbial Community Structure in Southwest China"

_microorganisms, 2023, doi:10.3390/microorganisms11112621_

Round 1
Reviewer 1 Report
The article is very interesting, but what is its uniqueness and difference from other articles on the same topic?
What is the originality of this article?
What is the point of Table 1 if more than 50% of this table is without information on options and deadlines?
Author Response
Q1: The article is very interesting, but what is its uniqueness and difference from other articles on the same topic? What is the originality of this article?
Response: Thank you for reading our manuscript with interest and providing positive comments. We think that our study found that different maize rotation patterns had significant effects on the physicochemical properties, enzyme activities, microbial biomass, and microbial community structure and diversity of rhizosphere soil. The comprehensive evaluation values of soil fertility for the different rotation patterns were ranked CG > Cir2 > CR > Cir3 > CC > Cir1. Our results indicated that NO3−-N, NH4+-N and MBN concentrations had greater impacts on soil bacterial communities, whereas NO3−-N, NH4+-N, AP, SOC and MBC concentrations had greater impacts on soil fungal communities. Our study provides empirical evidence supporting the introduction of garlic and annual ryegrass in local maize rotation patterns and extending the rotation period of annual ryegrass plantations to two years in subtropical mountainous areas of Southwest China.
Q2: What is the point of Table 1 if more than 50% of this table is without information on options and deadlines?
Response: We apologize for the table to misunderstand. In Table 1,‘---‘ represent the crop rotation test has not started in this plot, which belongs to maize monoculture. (please see lines 123-124)
Reviewer 2 Report
1. Authors must shorten the abstract. Add specific information about the impact of maize-crop rotation on the microbial community structure.
2. Avoid using “we” in the text.
3. Section 2.5 caused confusion. The V3/V4 regions of the 16S rDNA: how were the authors able to estimate the composition of fungal communities?
4. There is no data on how many sequences were obtained, what quality they were, how the bioinformation data was processed, how many OTUs were obtained.
5. The description of changes in the microbial community is given very superficially, only on the phylum level; the authors ignore seasonal changes in the community and do not consider many other factors. Data on the genus level must be presented and discussed thoroughly.
6. In the discussion of microbial communities, there is a description with a minimal discussion of the results obtained; it is necessary to refine, give reasoning, what changes in the composition of communities were significant, and what conclusions the authors ultimately draw from this.
6. "Conclusions" must be specific.
Considering the main scope of the journal, I conclude that the article cannot be considered for publication in Microorganisms in the current form.
-
Author Response
- Authors must shorten the abstract. Add specific information about the impact of maize-crop rotation on the microbial community structure.
Response: In the revised manuscript, the abstract had been brevity. And we have discussed our results systematically and in depth, and have focused more closely on the topic of the effects of maize-crop rotation on soil properties and microbial communities. (please see Abstract and Discussion)
- Avoid using “we” in the text.
Response: It was corrected in the revised manuscript.
- Section 2.5 caused confusion. The V3/V4 regions of the 16S rDNA: how were the authors able to estimate the composition of fungal communities?
Response: In the revised manuscript, this part was amended to read: " The ITS2 region of the ITS rDNA sequence was amplified from the genomic DNA. The fun-gal primer sequences were ITS3_KYO2 (GATGAAGAACGYAGYRAA) and ITS4 (TCCTCCGCTTATTGATATGC). The detailed conditions for PCR and amplicon se-quencing were described in our recent study (Wang et al., 2021). PCR reactions were performed in triplicate using an Applied Biosystems ProFlex 2 × 96-well PCR instrument (9902, ABI, USA). The sequencing was performed in Hiseq2500 Illumina system with PE250 mode by Guangzhou Genedenovo Co., Ltd., China." (please see lines 158-200)
- There is no data on how many sequences were obtained, what quality they were, how the bioinformation data was processed, how many OTUs were obtained.
Response: In the revised manuscript, we added a description of the results of the OTUs. " A total of 2,234,017 bacterial sequences and 2,366,741 fungal sequences were ob-tained. High-quality sequences constituted 91.26% of the original bacterial sequences and 91.09% of the original fungal sequences (Table.S1). These proportions suggest that the sampling and sequencing depths were satisfactory. Additionally, for bacterial OTUs, the ranges obtained were 3,978-4,947 in CC, 3,909-4,600 in CG, 4,624-5,560 in CR, 3,569-4,775 in Cir1, 2,388-4,071 in Cir2, and 4,566-5,870 in Cir3 (Table.S1). For fungal OTUs, the ranges were 766-895 in CC, 707-826 in CG, 746-817 in CR, 725-766 in Cir1, 644-804 in Cir2, and 695-989 in Cir3 (Table.S1)." (please see lines 271-278)
- The description of changes in the microbial community is given very superficially, only on the phylum level; the authors ignore seasonal changes in the community and do not consider many other factors. Data on the genus level must be presented and discussed thoroughly.
Response: Thank you for your very valuable comments. We also present and discuss the results at the generic level in the newly submitted manuscript. (please see lines 506-521, Table S2 and Table S3)
- In the discussion of microbial communities, there is a description with a minimal discussion of the results obtained; it is necessary to refine, give reasoning, what changes in the composition of communities were significant, and what conclusions the authors ultimately draw from this.
Response: In the revised version of the manuscript, we have discussed our results systematically and in depth, and have focused more closely on the topic of the effects of maize-crop rotation on soil properties and microbial communities in the expectation of providing some new perspectives. (please see 527-535)
- "Conclusions" must be specific.
Response: In the revised manuscript, we rewrote the conclusion to make it more specific. (please see lines 546-556)
Reviewer 3 Report
The content of the paper is interesting and is suitable to Microorganisms however I consider that the ms should be modified before publication. The aims of the paper a) to determine the effects of different maize rotation patterns on soil physicochemical properties, enzyme activity, microbial biomass and microbial community structure and diversity; and b) to explore the mechanisms driving soil microbial community structure effects on soil nutrient concentrations under different rotation patterns and determine the major environmental factors that affect soil microbial community structure, are very ambitious.
The main criticism is that I have serious doubts about if the experimental design is adequate for fulfil the aims of the paper.
Information concerning the experimental design and soil sampling is scarce (e.g. previous history of experimental area, homogeneity, soil management practices during the crop rotation systems-residues incorporation, tillage practices, times of crop rotations-oct 2028 Apr. 2029???-, collection of the rhizosphere samples, representative soil samples, etc.). The number of soil physical and chemical properties is reduced (6) and I have serious doubts that they can be used to evaluate soil fertility. The study includes a reduced number of samples that did not allow to analyse the scarce data with the selected statistical analysis. The matrix data used in the analysis is not adequate and the number of soil samples used are not indicated. They used only 1 soil with 6 different crop rotation systems (CC, CG, CR, Cir1, Cir2 and Cir3). If I understood correctly 4 plots of each crop system were used, which allows us to explore the spatial variation of the soil properties under field conditions but only the mean value of these replicates can be used for most statistical analyses (e.g. correlations, bacterial and fungal data, etc..). Therefore only 6 soil samples instead of 24 (6 treatments x 4 replicates????) can be included in the analyses.
I have also doubts about experimental data which include marked and inconsistent changes in some physicochemical (1 unit pH changes, marked changes SOC, TN and TP, see table 2) and in biochemical properties (marked changes of 1.5-4 fold times in microbial C, N and P) after only 3 years crop systems management. Marked changes in microbial biomass are not expected since bacteria (prokaryote) are dominant rhizosphere zone and its contribution to microbial biomass are much lesser than that of fungi (eukaryote) (see figure 3). In addition, microbial biomass N values can not be lower than microbial biomass P. Data should be analysed with precaution since significant differences among treatments can not be relevant under field conditions (spatial and temporal data variation). Likewise, correlations have to be examined with precaution (meaning of the correlations between microbial biomass values and microbial taxa parameters???). Results should be shortened and focused in the results obtained in the paper.
To sum up, I consider that the paper cannot be acceptable for publication in the present form.
Author Response
The content of the paper is interesting and is suitable to Microorganisms however I consider that the ms should be modified before publication. The aims of the paper a) to determine the effects of different maize rotation patterns on soil physicochemical properties, enzyme activity, microbial biomass and microbial community structure and diversity; and b) to explore the mechanisms driving soil microbial community structure effects on soil nutrient concentrations under different rotation patterns and determine the major environmental factors that affect soil microbial community structure, are very ambitious.
The main criticism is that I have serious doubts about if the experimental design is adequate for fulfil the aims of the paper.
Information concerning the experimental design and soil sampling is scarce (e.g. previous history of experimental area, homogeneity, soil management practices during the crop rotation systems-residues incorporation, tillage practices, times of crop rotations-oct 2028 Apr. 2029???-, collection of the rhizosphere samples, representative soil samples, etc.). The number of soil physical and chemical properties is reduced (6) and I have serious doubts that they can be used to evaluate soil fertility. The study includes a reduced number of samples that did not allow to analyse the scarce data with the selected statistical analysis. The matrix data used in the analysis is not adequate and the number of soil samples used are not indicated. They used only 1 soil with 6 different crop rotation systems (CC, CG, CR, Cir1, Cir2 and Cir3). If I understood correctly 4 plots of each crop system were used, which allows us to explore the spatial variation of the soil properties under field conditions but only the mean value of these replicates can be used for most statistical analyses (e.g. correlations, bacterial and fungal data, etc..). Therefore only 6 soil samples instead of 24 (6 treatments x 4 replicates????) can be included in the analyses.
I have also doubts about experimental data which include marked and inconsistent changes in some physicochemical (1 unit pH changes, marked changes SOC, TN and TP, see table 2) and in biochemical properties (marked changes of 1.5-4 fold times in microbial C, N and P) after only 3 years crop systems management. Marked changes in microbial biomass are not expected since bacteria (prokaryote) are dominant rhizosphere zone and its contribution to microbial biomass are much lesser than that of fungi (eukaryote) (see figure 3). In addition, microbial biomass N values can not be lower than microbial biomass P. Data should be analysed with precaution since significant differences among treatments can not be relevant under field conditions (spatial and temporal data variation). Likewise, correlations have to be examined with precaution (meaning of the correlations between microbial biomass values and microbial taxa parameters???). Results should be shortened and focused in the results obtained in the paper.
Response: Thank you for your comments concerning our manuscript. The comments were all valuable and very helpful for revising and improving our paper, as well as communicating the guiding significance of our research. In the revised manuscript, we made extensive changes to the Materials and Methods sections to make the descriptions more explicit. And we have discussed our results systematically and in depth, and have focused more closely on the topic of the effects of maize-crop rotation on soil properties and microbial communities in the expectation of providing some new perspectives.
Round 2
Reviewer 2 Report
The authors worked on the article and made many changes, but there are certain points that, it seems to me, require adjustment:
1. Please shorten the abstract and provide specific data on the microbial diversity of soils (main characteristics of microbial diversity, 1-2 sentences).
2. Still, the pronoun WE appears in the work; please correct it.
3. Do not break tables into different pages.
4. Material about the most significant genera of fungi and bacteria must be reflected in the results, since the reader becomes acquainted with them only in the discussion section.
5. It is necessary to include a section regarding supplementary information in the article itself; there are no links to supplementary tables.
-
Author Response
Response to the Reviewer
- Please shorten the abstract and provide specific data on the microbial diversity of soils (main characteristics of microbial diversity, 1-2 sentences).
Response: The abstract section has been streamlined and provided specific data on the microbial diversity of soils in the revised manuscript (please see line23-33).
- Still, the pronoun WEappears in the work; please correct it.
Response: The pronoun "WE" had been removed in the revised manuscript.
- Do not break tables into different pages.
Response: Adjustments have been made.
- Material about the most significant genera of fungi and bacteria must be reflected in the results, since the reader becomes acquainted with them only in the discussion section.
Response: The genera of fungi and bacteria had been exhibited in the result section in the revised manuscript. (please see line 315-324)
- It is necessary to include a section regarding supplementary information in the article itself; there are no links to supplementary tables.
Response: All supplementary material has been uploaded to Microorganisms's editorial system.